# Dissemination of Clinical *Acinetobacter baumannii* Isolate to Hospital Environment during the COVID-19 Pandemic

**DOI:** 10.3390/pathogens12030410

**Published:** 2023-03-03

**Authors:** Emina Pustijanac, Jasna Hrenović, Mirna Vranić-Ladavac, Martina Močenić, Natalie Karčić, Lorena Lazarić Stefanović, Irena Hrstić, Jasenka Lončarić, Martina Šeruga Musić, Marina Drčelić, Dijana Majstorović, Ines Kovačić

**Affiliations:** 1Faculty of Natural Sciences, Juraj Dobrila University of Pula, 52100 Pula, Croatia; 2Department of Biology, Faculty of Science, University of Zagreb, 10000 Zagreb, Croatia; 3Public Health Institute of Istria County, 52100 Pula, Croatia; 4General Hospital Pula, 52100 Pula, Croatia; 5Faculty of Medicine, Juraj Dobrila University of Pula, 52100 Pula, Croatia; 6Faculty of Educational Sciences, Juraj Dobrila University of Pula, 52100 Pula, Croatia

**Keywords:** *Acinetobacter baumannii*, air conditioner, COVID-19, environment, hospital, resistance

## Abstract

The aim of this study was to find the source of *Acinetobacter baumannii* in the intensive care unit (ICU) after an outbreak during the coronavirus disease 2019 (COVID-19) pandemic, as there was no *A. baumannii* detected on usually screened susceptible surfaces. The screening of the ICU environment was done in April 2021 when eleven different samples were taken. One *A. baumannii* isolate was recovered from the air conditioner and was compared with four clinical *A. baumannii* isolates obtained from patients hospitalized in January 2021. Isolates were confirmed by matrix-assisted laser desorption ionization-time of flight mass spectrometry (MALDI-TOF MS), minimum inhibitory concentrations (MICs) were determined, and the multilocus sequence typing (MLST) was performed. The molecular identification of *A. baumannii* isolates as ST208, the presence of the same *bla*_OXA-23_ carbapenemase gene, and the same antibiotic susceptibility profile suggest that the isolate recovered from the air conditioner is the same as the isolates recovered from hospitalized patients. The environmental isolate was recovered three months later than the clinical isolates, emphasizing the ability of *A. baumannii* to survive on dry abiotic surfaces. The air conditioner in the clinical environment is an important but undoubtedly neglected source of *A. baumannii* outbreaks, hence, frequent disinfection of hospital air conditioners with appropriate disinfectants is mandatory to mitigate the circulation of *A. baumannii* between patients and the hospital environment.

## 1. Introduction

*Acinetobacter baumannii* is a Gram-negative aerobic coccobacillus perceived as an emerging opportunistic pathogen often associated with multidrug-resistant infections, especially ventilator-associated pneumonia, in vulnerable patients in intensive care units (ICUs) [1,2]. They present the ability to form biofilms and to rapidly acquire resistance determinants to a wide range of antibacterial agents that enable and improve survival and spread in the hospital environment [1,3].

The greatest risk for pathogen transmission within the hospital environment is the activities of healthcare workers, such as hand and glove contamination after contact with the patient, even though the hospital environment itself is an increasingly important reservoir of *A. baumannii* [4,5,6,7].

Multidrug-resistant *A. baumannii* has been found on bed curtains, laryngoscopes, patient lifting equipment, anti-decubitus mattresses, pillows, medical charts, bedside cabinets, door handles, mops, sinks, floors, keyboards, telephones, ventilation grates, and equipment for mechanical ventilation and aspiration [4,5,6].

*A. baumannii* isolates have been seen to survive up to 60 days on different dry inanimate materials such as filter paper, Formica, glass, cotton, and metal [7,8,9,10,11,12]. Moreover, one multidrug-resistant isolate survived on a dry laboratory coat for more than 90 days [13]. The long-term persistence on dry surfaces of the hospital environment confirms the source of the spread of hospital outbreaks. In vitro studies have shown that *A. baumannii* can endure a wide range of pH values and temperatures in different water media, regardless of the availability of dissolved oxygen [14,15]. The aforementioned certainly raises a serious public health concern, and given that *A. baumannii* biofilms on glass and ceramic have reduced susceptibility to disinfectants, the disquietude becomes even greater [16].

During the coronavirus disease 2019 (COVID-19) pandemic, *A. baumannii* was recognized as a frequently implicated pathogen that contributed to outbreaks of multidrug-resistant organisms in both ICU and non-ICU units [17,18]. Numerous factors, such as the appropriate use of personal protective equipment, adherence to hand hygiene protocols, appropriate storage of personal protective equipment, and responsible antibiotic use, may all contribute to outbreaks of multi-drug-resistant organisms in hospital settings. However, these factors can potentially be modified to prevent the transmission of these organisms in healthcare facilities [17]. Throughout the COVID-19 pandemic, healthcare workers faced challenges in implementing standard precautions because they had to wear the same equipment for extended periods, prioritize the continuity of patient care, and concentrate on safeguarding themselves rather than preventing the spread of bacteria in the wards. The combination of overcrowded wards, understaffing, and a shortage of properly trained infection control professionals, along with all the aforementioned factors, resulted in the identification of a significant amount of confirmed and potentially more unidentified outbreaks of *A. baumannii* during the COVID-19 pandemic [17,19,20,21].

The aim of this study was to find the source of *A. baumannii* in the ICU after the outbreak, as no *A. baumannii* was detected on the usually screened susceptible surfaces. In this study, we report the first evidence of clinically relevant *A. baumannii* disseminated from patient to air conditioner in the ICU environment during the COVID-19 outbreak.

## 2. Materials and Methods

### 2.1. Isolation of A. baumannii

The study was performed at General Hospital Pula, a 427-bed hospital in the city of Pula, Croatia. From 1 December 2020, until 6 June 2021, an eight-bed intensive care unit (ICU) for COVID-19 patients was established at the former location of this hospital. During this period, 67 patients with severe COVID-19 symptoms were hospitalized in the ICU, and 29 of them acquired a co-infection with *A. baumannii*. As a critical aspect of the infection control policy, surveillance cultures were obtained from all 67 patients before their admission to the ICU and then on a weekly basis following their admission, in accordance with hospital protocol. In January 2021, six patients acquired *A. baumannii* infection. In February and March, the number of patients decreased, and there were no *A. baumannii* isolated from ICU patients. In spring 2021 the number of severe COVID-19 patients increased again. Co-infection with *A. baumannii* was detected during April and May in 15 and 8 patients, respectively.

The screening of the ICU environment for *A. baumannii* was done in April 2021. Eleven samples were collected from the ICU environment. The type and origin of the samples were different and consisted of nine swabs and two material samples (Table 1). Additionally, the hospital conducted an annual environmental screening from 2015 to 2020, which covered the area where a COVID-19 ICU was established in 2020. This resulted in the availability of 110 additional environmental samples with varying origins and sampling outcomes, but *A. baumannii* was not detected in any of them.

One *A. baumannii* isolate was recovered from the air conditioner. The sample was a swab from the plastic exhaust fan of the indoor unit. The air conditioner was located at a height of 2.5 m in the ICU room with a total 3.5 m height and a surface area of 40 m^2^.

Four clinical *A. baumannii* isolates obtained from patients hospitalized in January 2021 were selected for comparison with the isolate recovered from the air conditioner (Table 2). The clinical isolates from January were the earliest ones identified in the outbreak and they were chosen because they had the greatest time interval with respect to the isolate retrieved from the air conditioner.

### 2.2. Identification of A. baumannii

The isolation and identification of isolates as the *A. baumannii* complex was performed by routine bacteriological techniques. Further confirmation of *A. baumannii* species was done by matrix-assisted laser desorption ionization-time of flight mass spectrometry MALDI-TOF MS (Bruker MALDI Biotyper, software version 5.0.2., Bremen, Germany) on cell extracts. In order to determine the genetic relatedness of *A. baumannii* isolates, multilocus sequence typing (MLST) was performed according to the Oxford MLST scheme and encompassing seven housekeeping genes (*gltA*, *gyrB*, *gdhB*, *recA*, *cpn60*, *gpi*, and *rpoD*). Primers and conditioners, as described at PubMSLT (http://pubmlst.org/abaumannii/; accessed on 6 April 2022) were used. Fragments amplified by polymerase chain reaction (PCR) (ProFlex™ 96-Well PCR System, Applied Biosystems, Foster City, CA, USA) were sequenced (commercial service GENEWIZ, Azenta Life Sciences, Griesheim, Germany) and edited using Geneious Prime software (http://www.geneious.com/; accessed on 6 April 2022). Allele sequences and profiles retrieved from the *A. baumannii* MLST website (http://pubmlst.org/perl/bigsdb/bigsdb.pl?db=pubmlst_abaumannii_oxford_seqdef; accessed on 7 May 2022) were used to determine the sequence type (ST) and international clonal lineage (IC).

### 2.3. Antibiotic Susceptibility Testing and Detection of Carbapenemase Genes

The susceptibility to carbapenems (meropenem and imipenem), fluoroquinolones (ciprofloxacin and levofloxacin), aminoglycosides (amikacin, gentamicin, and tobramycin), and trimethoprim/sulfamethoxazole were evaluated by the determination of the minimum inhibitory concentration (MIC) values obtained with the Vitek2 system (BioMérieux, Marcy I′Etoile, France). Colistin resistance was tested by the broth microdilution test (Mikrolatest, Erba Lachema, Brno, Czech Republic). The MICs were interpreted according to the recent European Committee on Antimicrobial Susceptibility Testing (EUCAST) breakpoints (v 12.0) for clinical isolates of *Acinetobacter* spp. (https://www.eucast.org/clinical_breakpoints/, accessed on 6 June 2022).

The presence of acquired genes of the *bla*_OXA_ lineage that encode OXA-type carbapenemases were searched to explain the molecular mechanism of carbapenem resistance. The multiplex polymerase chain reaction (PCR) with specific primers for *bla*_OXA-40-like_, *bla*_OXA-23-like_, and *bla*_OXA-58-like_ genes was performed according to Woodford et al. [22]. Obtained amplified fragments of *bla*_OXA-23-like_ genes were sequenced, edited, and analyzed according to the protocol described for MLST determination.

## 3. Results

Regular surveillance cultures were conducted on 67 patients with severe COVID-19 symptoms. The percentage of patients in the ICU department who were infected with *A. baumannii* during the study period was 43% (Table 3). It is not surprising that all isolates were resistant to carbapenems, as carbapenem resistance in *A. baumannii* in Croatia had reached 99.5%, according to the Surveillance Atlas of Infectious Diseases of the European Centre for Disease Prevention and Control (https://atlas.ecdc.europa.eu/public/index.aspx?Dataset=27&HealthTopic=4, accessed on 4 February 2023).

From 2015 to 2020, the hospital environment underwent annual screening, and 110 additional samples were obtained. Nevertheless, *A. baumannii* was not detected in any of the samples collected.

The isolate recovered from the air conditioner, together with four selected clinical isolates, were accurately identified by MALDI-TOF MS as *A. baumannii*. The MLST revealed that all isolates had the same allele profile and belonged to the ST208 within IC2 (Table 2). The IC2 clonal lineage of *A. baumannii* is the most widespread among the eight international lineages and is accountable for most of the outbreaks [23,24].

Each one of the five isolates was highly resistant to carbapenems, fluoroquinolones, and aminoglycosides, but remained sensitive to colistin and trimethoprim-sulfamethoxazole (Table 4).

Therefore, isolates could be classified as multidrug-resistant (MDR) according to Magiorakos et al. [25]. All isolates harbored the identical acquired *bla*_OXA-23_ carbapenemase gene spread worldwide, which showed 100% sequence ID with numerous sequences deposited in GenBank (for example AJ132105, CP076802, CP045645).

## 4. Discussion

The molecular identification of *A. baumannii* isolates as ST208, the presence of the same acquired *bla*_OXA-23_ carbapenemase gene, and the same antibiotic susceptibility profile suggest that the isolate recovered from the air conditioner is the same as the isolates recovered from hospitalized patients. The *bla*_OXA-23_ is commonly present in the *A. baumannii* isolates belonging to the IC2 in Croatia [26]. Since it is an acquired carbapenemase gene, its presence in hospital isolates indicates a risk of dissemination of resistance to this last-resort antibiotic [22]. The isolate obtained from the air conditioner displayed varying MIC SXT values, which may have resulted from adaptation to unfavorable environmental conditions. This may be because it was obtained much later than the clinical isolates from the plastic indoor unit of the air conditioner. Bacteria are capable of retaining infection-promoting factors even under stressful conditions, such as prolonged periods of desiccation [11,16]. As a result, co-infection of COVID-19 patients hospitalized in the ICU with this strain could greatly increase the risk factors and complicate the hospital course of patients. Studies during the pandemic have emphasized that co-infection of COVID-19 patients with *A. baumannii* heightens the likelihood of developing difficult-to-treat infections [11,21].

The air conditioner in the ICU environment where the patients with severe COVID-19 symptoms were hospitalized worked continuously throughout the day and night. It was merely turned off during February and March due to the decreased number of patients. In April, the number of severe COVID-19 patients increased, hence the air conditioner was turned on again and was neither cleaned nor disinfected. The air conditioners in the hospital were regularly mechanically cleaned, nevertheless, they had never been disinfected until after the detection of the *A. baumannii* isolates.

In a study conducted at General Hospital Pula from 2012 to 2014, various clones of *A. baumannii* were identified. These clones were predominantly obtained from the ICUs and other hospital departments such as surgery, internal medicine, neurology, and otorhinolaryngology [26]. However, none of these clones were found to be of the ST208 type, which highlights that the outbreak that recently emerged occurred in a newly established ICU. Most probably, the spread of *A. baumannii* belonging to ST208 to the air conditioner occurred via aerosols from the respiratory tract of the first co-infected patient in this ICU (isolate no. 113). The isolate from the air conditioner was recovered three months later than the clinical isolate no. 113. This is explained by the ability of *A. baumannii* to survive on dry abiotic surfaces for several months [11,13,27]. After attachment to the abiotic surface, *A. baumannii* survives in biofilm as viable or even dormant cells [6,11,15,27]. Certain strains of *A. baumannii* can maintain or even increase their ability to produce biofilms after rehydration and prolonged starvation [6,11]. In addition, biofilms are capable of forming on both living and non-living surfaces and provide protection to bacteria from environmental damage such as host responses, antibiotics, cleansers, and disinfectants [9]. The production of biofilms and their ability to withstand desiccation can heighten the probability of the establishment and persistence of *A. baumannii* in hospital environments, as well as increase the risk of acquiring antimicrobial resistance, and causing nosocomial infections and outbreaks [6,9,10]

When *A. baumannii* colonizes an air conditioner, the cells could spread through the hospital room via airflow. This represents a direct source of further infection for immunocompromised patients and even hospital personnel [28,29,30]. Regular service and frequent disinfection of air conditioners in hospital rooms are some of the ways to mitigate the dissemination of clinical *A. baumannii* to the hospital environment and vice versa. Here, special attention should be given to the resistance of *A. baumannii* biofilms to commercial disinfectants [16]. The need for the identification of novel disinfectant compounds to treat resistant pathogens, such as *A. baumannii*, has been recognized and emphasized [31,32].

It can be concluded that the air conditioner in the clinical environment is an important but undoubtedly neglected source of *A. baumannii* outbreaks. Frequent disinfection of hospital air conditioners with appropriate disinfectants is mandatory to mitigate the circulation of *A. baumannii* between patients and the hospital environment.

## Figures and Tables

**Table 1 pathogens-12-00410-t001:** Type, origin of the sample, and colony forming units (CFU) of detected bacteria during the screening of the ICU environment in General hospital Pula occurring in April 2021.

Origin of the Sample	CFU/Sample	Detected Bacteria
Powder for gloves in a plastic cup	50	*Enterococcus faecalis*
Powder for gloves—bulk	0	sterile
Swab from faucet in front of the ICU	105	*Pseudomonas aeruginosa*
Swab from faucet in isolation box	105	*Pseudomonas aeruginosa*
Swab from faucet in the middle box	50	*Pseudomonas aeruginosa*
Swab from patient monitor 1	0	sterile
Swab from patient monitor 2	0	sterile
Swab from faucet in front of the clinic	0	sterile
Swab from water jug in isolation box	0	sterile
Swab from stainless steel table for therapy preparation	0	sterile

**Table 2 pathogens-12-00410-t002:** Origin, date of isolation, and molecular identification of *A. baumannii* * isolates from general hospital Pula.

Isolate	Origin	Date of Isolation	Sequence Type (ST)
1524	air conditioner	7 April 2021	208
113	bronchial aspirate	5 January 2021	208
737	tracheal aspirate	18 January 2021	208
798	bronchoalveolar aspirate	19 January 2021	208
809	tracheal aspirate	19 January 2021	208

* MALDI-TOF MS provided high-confidence species identification for *A. baumannii* with score values from 2.10 to 2.36.

**Table 3 pathogens-12-00410-t003:** Percentage of patients infected with *A. baumannii* in the ICU department during the study period.

The Month and the Year	Number of Patients	Number of Positive Patients	Percentage
December 2020	10	0	0%
January 2021	9	6	67%
February 2021	1	0	0%
March 2021	9	0	0%
April 2021	22	15	68%
May 2021	16	8	50%
All together	67	29	43%

**Table 4 pathogens-12-00410-t004:** Antibiotic * susceptibility profile of *A. baumannii* isolates from general hospital Pula.

Isolate	MIC Values of Antibiotics (mg/L)								
	IPM	MEM	CIP	LVX	AMK	GEN	TOB	CST	SXT
1524	≥16 R	≥16 R	≥4 R	>2 R	≥64 R	≥16 R	≥16 R	0.50	1.0
113	≥16 R	≥16 R	≥4 R	>2 R	≥64 R	≥16 R	≥16 R	0.50	0.5
737	≥16 R	≥16 R	≥4 R	>2 R	≥64 R	≥16 R	≥16 R	0.50	0.5
798	≥16 R	≥16 R	≥4 R	>2 R	≥64 R	≥16 R	≥16 R	0.50	0.5
809	≥16 R	≥16 R	≥4 R	>2 R	≥64 R	≥16 R	≥16 R	0.50	0.5

* MEM, meropenem; IPM, imipenem; CIP, ciprofloxacin; LVX, levofloxacin; AMK, amikacin; GEN, gentamicin; TOB, tobramycin; CST, colistin; and SXT, trimethoprim-sulfamethoxazole. R resistant according to EUCAST criteria. All isolates harbored the identical acquired *bla*_OXA-23_ carbapenemase gene.

## Data Availability

The data supporting the conclusions of this article are provided within the article. The original data in the present study are available from the corresponding authors.

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
