# Peer review of "Dissemination of Clinical Acinetobacter baumannii Isolate to Hospital Environment during the COVID-19 Pandemic"

_pathogens, 2023, doi:10.3390/pathogens12030410_

Round 1
Reviewer 1 Report
Authors show here an outbreak caused by a cross-contamination from an A/C machine with A. baumannii in a COVID unit in short-term during the pandemic.
I found the work well written but more references should be included, as well as larger discussion to highlight the significance of the paper.
I included some major and minor comments to be addressed.
Major:
- In subsection 2.1. you describe the number of patients that were co-infected with A. baumannii. The rest of the people were not tested or were negative? Could you please specify (in case they were tested for different symptoms, for example, or if everyone was tested the percentage of positive samples)? Please explain the rationale for picking those patients and include in the result section the data.
- Subsection 2.1. You describe 4 clinical samples and 1 environmentally isolated one. Why did you choose only those ones? The rest are not characterized or were not linked to the outbreak? All of them should be at least MLST tested, and included. Again, please explain the rationale for picking those strains only and include in the result section the data.
Line 107: "recent" EUCAST breakpoint. Which version? Please specify and include the date for the webpage accession.
- Line 119: "IC2". Please, explain the meaning of the clones and include references.
- Line 128: According to Magiorakos et al., 2012 for considering Acinetobacter XDR should be resistant to more than 1 agent in all but 2 categories. Since nor tetracyclines or cephalosporins were tested, you can only conclude that they are MDR (doi: 10.1111/j.1469-0691.2011.03570.x).
- Line 134: Since they are all sensitive to SXT, they show different MIC value. Since you conclude that is the same strain, how would you justify that?
- Line 143-145: You mention another study performed in the same facilities, could you please compare the results and the places in which the strains were previously found?
Extra references:
- Lines 43-46: doi: 10.1186/s40560-015-0120-5; 10.1016/S0195-6701(07)60015-2
- Lines 47-48: up to 60 days survival different strains, doi: 10.1371/journal.pone.0201961
More references are also needed in the discussion, specially in the paragraph lines 151-156 and lines 148-150. Please compare your results with the ones found in other places and explain briefly the relevance of the biofilm or the dormant or VBNC cell stages.
Minor:
- Line 63: From December 1st 2020 until June 6th 2021
- Line 110: blaOXA
- Line 188: A. baumannii
- Line 129: blaOXA-23
Reviewer 2 Report
In this study, authors performed the one-time point screening of specific intensive care unit environment sites. Acinetobacter baumannii of ST208 was recovered from the ICU air conditioner, and found ST208 A. baumannii in the COVID-19 patient samples previously admitted to the same ICU. Highlighting the role of the air conditioner as a potential source of A. baumannii nosocomial infection is important for the infection control unit. However, limited sampling was performed in this study, and only a few isolates were characterized, making it hard to understand the contamination of the level of A. baumannii in the respective healthcare facility and its role in hospital-acquired infection.
1- "The aim of this study was to find the source of Acinetobacter baumannii in the intensive care unit (ICU) after the outbreak, as there was no A. baumannii detected on usually screened susceptible surfaces." Any previous work was done before COVID-19 on the same site. The dissemination of A. baumannii detection to the hospital environment during the COVID-19 pandemic is required further discussion and justification. Survival of A. baumannii on a biotic surface in the hospital is well-known and commonly reported in the literature from ICU sites.
2- The brief methodology is missing in the abstract for the experiments and analysis performed in this study.
3- The survival capacity of A. baumannii in the environment has been explained in the introduction. The introduction needs further improvement on the role of the COVID-19 pandemic and detection of A. baumannii and the status of hospital-acquired infection from A. baumannii.
4. Total of 29 A. baumannii isolates were recovered from COVID-19 patients during the study period, and characterization was performed for only four strains, making it hard to conclude the situation of A. baumannii in the respective healthcare facility.
4- A limited number of one-time sampling was performed mentioned in Table 1. Potential other surfaces were missed that could be a source of nosocomial pathogens in the ICU patients.
5- Page 2, line 76. It would be better to avoid detail of the manufacturer of the air conditioner.
6- Page 2, line 79. In January 2021, six patients acquired A. baumannii infection. Please explain why four isolates were selected for comparison. Moreover, the authors mentioned, "Co-infection with A. baumannii was detected during April and May in 15 and 8 patients, respectively". It was important to include those cases as they were collected almost in the same time period of the environment screening of the ICU.
7- Page 3, line 114. Obtained amplified fragments of blaOXA-23-like genes were sequenced, edited, and analysed as described above. No information is there above about the blaOXA-23 analysis.
8. Need to expand the discussion with more relevant literature about the important finding of this study.
Reviewer 3 Report
Dear editors,
The submitted manuscript does not describe any new techniques or/and innovative approaches.
However, the research addresses A. baumannii, raising awareness of a species of great public health importance.
Round 2
Reviewer 1 Report
The authors have answered all my previous comments and suggestions, increasing in my opinion the quality and significance of the paper, for which I accept the manuscript in its current version.
Reviewer 2 Report
The authors have addressed several comments in the revised manuscript. The limited sampling and one-time point analysis significantly reduced the quality of the overall study.
